# A MINIMALIST APPROACH FOR EXPLORING TRANSFORMER ROBUSTNESS TO IN-DISTRIBUTION AND OUT-OF-DISTRIBUTION SAMPLES

## ABSTRACT

Despite their strong performance across tasks, large language models (LLMs) still have limitations in their ability to generalize. Recent studies show that even state-of-the-art LLMs exhibit significant accuracy fluctuations when evaluated on superficially modified versions of the same benchmarks, suggesting potential gaps in their ability to generalize. We argue that current evaluation methods, which rely heavily on large Transformer-based models trained on massive and often opaque datasets, often make it difficult to disentangle whether limitations arise from architecture, data coverage, or other factors. While addressing this question in full requires considerable computational resources, we propose a cost-effective, preliminary investigation. Our approach involves training a tiny Transformer-based decoder-only language model (with tens of millions of parameters) from scratch on a custom code reasoning task. To train this model, we generate data using a synthetic data generation tool that allows precise control over data distribution and volume. We perform multiple experiments using our framework to study the in-distribution and out-of-distribution robustness of these models, revealing their behavior under controlled settings.

## 1 INTRODUCTION

Large language models (LLMs) can be brittle to seemingly small, meaning-preserving input changes. For example, Mirzadeh et al. (2024) show that state-of-the-art models, despite strong scores on GSM8K (Cobbe et al., 2021), suffer sizable drops on slightly altered versions of the same problems. Similar fragility to minor perturbations (typos, synonyms, rephrasings) has been widely documented in NLP (Moradi & Samwald, 2021), and recent analyses indicate that even advanced LLMs may rely on superficial token biases that flip decisions under semantics-preserving edits (Jiang et al., 2024). In the *code* domain, related robustness limitations appear under semantics-preserving mutations (renamings, refactorings, neutral insertions), where neural code models and code-specific LLMs can change predictions despite unchanged program semantics (Henkel et al., 2022; Bielik & Vechev, 2020; Orvalho & Kwiatkowska, 2025). Beyond in-distribution (ID) semantics-preserving edits, Language Models (LMs) also struggle under out-of-distribution (OOD) shifts, especially compositional and structural changes, where scaling data and model size often yield limited gains (Keysers et al., 2020; Tsarkov et al., 2021; Anil et al., 2022).

While it is established that (i) LMs are sensitive to ID semantics-preserving changes and (ii) OOD generalization remains challenging, key questions persist. *How far can data scaling alone improve ID robustness?* Evidence suggests benefits, but it remains unclear whether data scaling alone can fully eliminate ID brittleness, or whether architectural/training changes are required. For OOD, we lack controlled studies that identify whether some axes (e.g., length increase vs. depth increase) remain difficult even under extensive data scaling. And *how much data coverage is sufficient (and necessary)* to unlock invariance and compositional generalization? Progress is hindered because many studies rely on large, opaque pre-training corpora, obscuring whether failures stem from data coverage, architectural limits, or other reasons. Answering these questions is difficult: it requires precise knowledge of the training distribution and of train–test shifts, and modern LLMs make controlled experimentation computationally prohibitive.

To address these challenges, we analyze robustness using tiny transformer-based language models trained from scratch on a custom code tracing task, which we call TinyTracing. In this task, the model reads a short program and writes a step-by-step *trace* of its execution, at each step indicating which line is being executed and what the current variable values are (Liu et al., 2023). The task is (1) simple enough for tiny models to learn; (2) rich in concepts (symbolic variables, arithmetic,

control flow); and (3) paired with tooling that allow data generation while controlling train/test distributions (details on the TinyTracing task and data generation tool in Sec. 2).

With this environment, we train a decoder-only transformer-based language model from scratch on synthetic TinyTracing datasets (the architecture and dataset are described in Sec. 3). We then evaluate ID/OOD robustness (Sec. 4) using a set of experiments: (i) ID robustness to semantics-preserving alterations (e.g., variable renaming, neutral line addition; Sec. 4.1); (ii) generalization to unseen samples under controlled coverage of variable-pair combinations in arithmetic expressions (Sec. 4.2); and (iii) OOD generalization to longer snippets and higher nesting depth (Sec. 4.3).

By synthesizing data, our framework gives *fine-grained control* over the training and test distributions: we can (a) isolate and hold out specific pattern classes, (b) tune coverage of those classes in the training set, and (c) orthogonally manipulate OOD axes (e.g., symbol coverage, sequence length, nesting depth) while keeping all other axes fixed. This control lets us ask questions that are hard to explore with opaque LLM pre-training. For example, we use our framework to explore the following questions, not yet explored in the literature: (i) whether increasing ID data alone eliminates brittleness to a set of semantic-preserving edits that we study; (ii) when does robustness to these semantics-preserving edits emerge during training, early in training (before loss convergence), or only later? (iii) whether some OOD axes remain difficult when increasing data alone (e.g., increasing code depth vs length); and (iv) how much coverage/minimal exposure to certain data patterns is enough to enable generalization?

**Summary of findings.** (1) Training on larger ID datasets largely diminishes ID brittleness to the semantics-preserving edits we study. (2) Surprisingly, robustness to these semantics-preserving edits emerges early in training, even when the accuracy of the model is still modest (at the 1st epoch). (3) Providing limited coverage of certain code patterns in the training set (e.g., variable-pair combinations used in expressions of the form `var = var1 op var2`) is enough to obtain high accuracy on unseen cases of those code patterns (e.g., new variable-pairs). (4) The same model in our tests partially extrapolates to longer sequences but fails on deeper nesting, showing qualitatively different difficulty across OOD axes. These results clarify cases when data scaling helps, where it might fall short, and quantify how limited coverage relates to certain types of compositional generalization.

In this paper, we study *decoder-only Transformer-based language models* trained from scratch on a *simple programming language* for a controlled code-tracing task. Therefore, our findings apply to this setting and do not necessarily generalize to other architectures, tasks, or ID/OOD types. Instead, they motivate the need for more *controlled* experiments where train/test distributions are known and manipulable, as a complementary path to large-scale evaluations for better understanding of LMs.

In summary, our **contributions** are as follows: **1)** We propose a cost-effective framework to study the robustness of transformer-based decoder-only language models in in-distribution and out-of-distribution settings; **2)** We show that training on larger ID datasets largely reduces ID brittleness to the semantics-preserving edits that we study; **3)** We show that robustness to the semantics-preserving edits that we study appears early in training, even when the accuracy of the model is still modest; **4)** We find that providing limited coverage of certain patterns is enough to obtain high accuracy on unseen cases of those patterns; 5) We release the full framework, data generation, training, and evaluation scripts, along with the datasets, to the community (will be released with the camera-ready paper to keep anonymity).

## 2 Designing the Probing Task and the Data Generation Tool

### 2.1 TinyTracing Task

We propose a controlled experimental setup in which models are trained to perform the task of *TinyTracing*. This task requires the model to take code as input and to generate its line-by-line execution steps (execution trace) by duplicating the code snippet at each execution step and annotating the duplicate with corresponding execution information, including the instruction pointer and the variable states (a depiction of the TinyTracing task in Figure 1). By adjusting the complexity of the programming language and code snippets used for tracing, the task can be adapted to fit the computational budget of users, allowing cost-effective experiments.

### 2.2 Data Generation Tool

To complete our experimentation setup, we need to be able to access arbitrary quantities of data with precisely controlled distributional characteristics. To this end, we propose a data generation

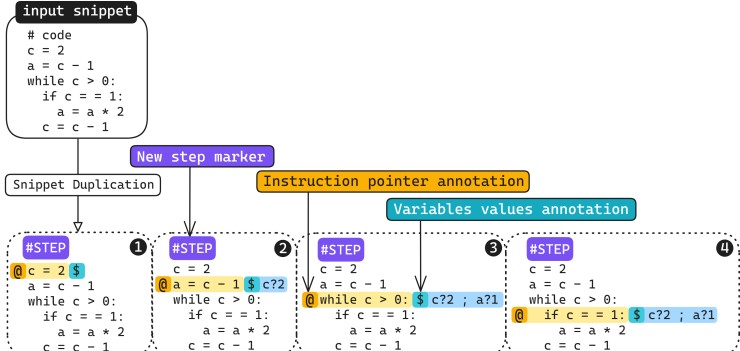

Figure 1: Illustration of the TinyTracing task. At each execution step (we show the first 4 steps), the entire input code snippet is duplicated and annotated with corresponding execution information: The instruction pointer value is represented by setting a special symbol (here @) to the left of the current line to be executed, and the variable states are presented as (key, value) pairs to the right of the current execution line. A special symbol (here #STEP) indicates the beginning of a new step (see Appendix A for a full example).

tool, referred to as **TinyTrace-Generator**. This tool enables the synthesis of random code snippets that are expressed in a subset of Python (or in Python) and that follow a certain data distribution (defined using a set of constraints on the Python language). The tool also allows the creation of the execution trace of these snippets according to the format of the TinyTracing task. TinyTrace-Generator is structured as a three-stage pipeline, as illustrated in Figure 2, and is described in greater detail in the following paragraphs.

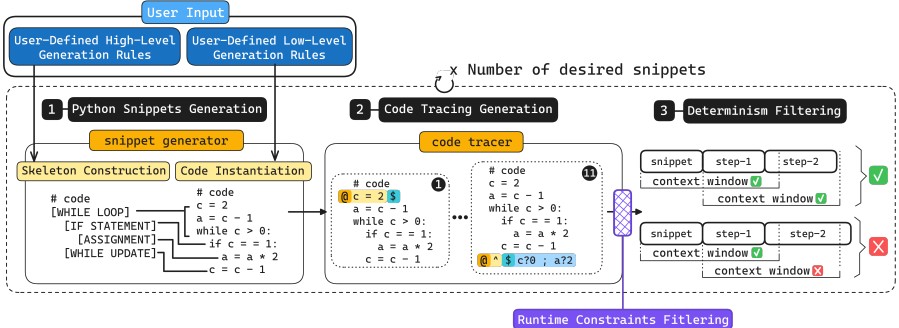

Figure 2: TinyTrace-Generator as a three-stage pipeline.

As depicted in the leftmost part of Figure 2, the first stage of TinyTrace-Generator produces code snippets for the task using an imperative, rules-based generator. It runs two coupled processes: skeleton construction (builds an abstract snippet with high-level keywords) and code instantiation (turns those keywords into concrete Python). Both are driven by user-defined procedural rules. Further details appear in Appendix B.

The second step of the pipeline involves generating the execution trace of code snippets created in the first stage. It leverages the debugging capabilities provided by the Python standard library to generate tracing. This setup not only enables straightforward access to key runtime information, such as the program counter and the state of variables, but also facilitates the implementation of runtime filtering barriers. These barriers serve to enforce constraints on runtime properties that cannot be determined during snippet generation. Depending on the specific requirements of the target experiment, such filters may include the detection of undesirable variable values during execution (e.g., excessively large integers), an unexpected number of loop iterations (including infinite loops), or simply the occurrence of runtime exceptions.

The third and final component of TinyTrace-Generator ensures that the generated tracing data allows deterministic inference. For each generated snippet, we check that every two consecutive steps (code duplicates) in its execution trace fit within the Transformer's context window after tokenization. This guarantees that during inference, the model has full access to the preceding execution step when generating a new one, preserving determinism. The generated snippets that do not meet this condition are filtered out. The application of this last stage can be considered optional, depending on the use of the data: For instance, for training, this may not be necessary as the random sampling of token batches from the training corpus is task-unaware, while for supervised fine-tuning or model evaluation, this stage is required so that inference becomes deterministic (Appendix C.5).

While the code generator supports the generation of loops and complex data structures, in this paper, we do not generate code that has loops or complex data structures.

# 3 MODEL, DATA AND EVALUATION METHODOLOGY

This section details the model architecture and the subset of Python that we use in our experiments. It also defines the evaluation protocol and metric used to measure model performance.

## 3.1 MODEL ARCHITECTURE

For our experiments, we employ a custom, decoder-only Transformer architecture, which we train from scratch. We use NanoGPT: a small-scale, open-source implementation inspired by GPT-2, developed by Karpathy (2022). The model is designed to be small enough for rapid experimentation while being sufficiently powerful to learn the TinyTracing task. The architecture comprises 12 layers, 16 attention heads, and an embedding dimension of 368, resulting in approximately 20 million trainable parameters. The context window is set to 512 tokens. Each Transformer block utilizes a pre-Layer Normalization, employing RMSNorm for normalization before the self-attention and feed-forward sub-layers. The feed-forward network uses a 4x expansion factor and a SiLU activation function. For tokenization, we use a custom tokenizer with a fixed 77-token vocabulary tailored to the TinyTracing syntax (identifiers a–z, keywords, operators, punctuation, tracing markers), with no learned subword segmentation. This keeps symbols atomic. More details about the tokenization and hyperparameters are in Appendices C.2, and C.3.

## 3.2 DATASET

To create a controlled and reproducible experimental environment, all datasets used in our experiments are synthesized using the TinyTrace-Generator under a set of constraints that define a subset of Python that we use in our experiments.

Given the limited capacity of our small Transformer models (and limited computational budget), we focus on a minimalist subset of the Python language, ensuring the TinyTracing task remains learnable. This subset is built around the following programming constructs: variable assignments, arithmetic operations, and conditionals. We enforce the following constraints to define this Python subset: **1) Syntactic Constraints:** The grammar and vocabulary of the generated code are intentionally limited. The grammar includes variable assignments, `if` conditional, and arithmetic expressions. Variable identifiers are restricted to a fixed set of 26 single lowercase letters (`a`–`z`). Arithmetic operations are limited to addition and subtraction (`+`, `-`), while conditional comparisons are limited to less-than and greater-than operators (`<`, `>`). The only supported data type is integers, and integer values are limited to be within the range [-99, 99]; **2) Structural Constraints:** The overall structure of the code snippets is also bounded. The total number of statements in any generated snippet is constrained to be between 5 and 10. Furthermore, the maximum nesting depth for control flow blocks (i.e., `if` statements) is limited to two; **3) Runtime Constraints:** Any generated snippet that results in a variable holding an integer value outside the range of [-99, 99] at any point during its execution is discarded. This filter ensures that the model only needs to learn representations for a bounded and manageable range of integer values.

We generate a dataset of 3 million code snippets that follow the distribution described above. We use this dataset (or sub-samples from this dataset sampled randomly), depending on the experiment. Unless stated otherwise, all experiments use the dataset generated under the distribution and language constraints defined previously; deviations (e.g., using longer code snippets) are explicitly noted where they occur. We use a hash-based deduplication method to ensure that code snippets in the dataset are unique across the training, validation, and test sets (detailed in Appendix C.4).

## 3.3 EVALUATION METHOD AND METRIC

We evaluate the model performance using **Exact Match Accuracy**. For each test sample, the model autoregressively generates the complete execution trace given the code snippet as a prompt. A generated trace is considered correct only if it is a character-for-character match to the ground truth.

## 3.4 MODEL TRAINING

We train our base model on a dataset of 3 million code snippets that follows the distribution of the Python subset described in Sec. 3. The total number of tokens is 933 million. We use $4 \times$ A100 GPUs, each with 80GB of memory, to train the model, and the training takes 1.5 hours (for 4 epochs). The model achieved an exact match accuracy of 100% on a test set of 1024 samples that follows the same distribution as the training set. In the experiments reported later, we intentionally vary dataset

size, training epochs, and dataset distributions depending on the experiment (e.g., smaller datasets require the use of more epochs to reach convergence). The base model reported in this section (and its accuracy) would help the reader contextualize the experiments and results reported later. Unless otherwise stated, all experimental models are architecturally identical to the base model; only study-specific knobs (e.g., dataset size, number of epochs) are varied (more details in Appendix C.1).

## 4 STUDYING IN-DISTRIBUTION AND OUT-OF-DISTRIBUTION ROBUSTNESS

### 4.1 ROBUSTNESS TO IN-DISTRIBUTION SEMANTICS-PRESERVING ALTERATIONS

We first study the behavior of our model when exposed to different in-distribution semantics-preserving alterations. The goal is to explore how the model reacts when exposed to altered versions of code snippets that it can already execute correctly, while ensuring that these altered versions remain within the distribution of the training data. We also want to study how this in-distribution robustness evolves as a function of training-set size and the amount of training.

Our in-distribution alteration operators are as follows: **(1) Variable renaming**: variable identifiers are replaced with other valid identifiers from the same distribution, preserving semantics. **(2) Comparison symmetry**: conditional comparisons are made equivalent by simultaneously swapping operands and flipping to the symmetric comparator (e.g., $a < b \mapsto b > a$). **(3) Addition commutativity**: swap the order of operands in addition expressions (e.g., $a + b \mapsto b + a$). This transformation is applied exclusively to addition, since subtraction is not commutative. **(4) Neutral operator**: a $+0$ or $-0$ is inserted in an assignment (e.g., $x = y \mapsto x = y + 0$), which preserves semantics; note that $0$ naturally occurs in the training data range $[-99, 99]$, so such edits are in-distribution. **(5) Neutral assignment**: a new assignment is inserted at a random location, defining a fresh variable name not used elsewhere, ensuring the execution trace is unaffected. An example illustrating these alterations is presented in Appendix D.

For our experiments, we create four distinct training datasets containing code snippets that follow the same distribution described in Sec. 3. These datasets include, respectively, 3M, 1.5M, 750K, and 375K snippets. On each of these four datasets, we train our model for eight epochs. We evaluate the model at the end of each epoch on a test set composed of 1024 snippets that follow the same distribution as the training dataset. We ensure that the test set snippets are not seen during training (more details about deduplication in Appendix C.4). Then, for each evaluation, we isolate the code snippets that were correctly executed. We then apply each of the 5 semantics-preserving operators on these correctly executed snippets (we apply one at a time). We obtain 5 different new test sets. Each set is the result of applying exactly one operator; operators do not co-occur within a single snippet or a single test set. After applying the semantics-preserving edits, the new altered codes remain within the distribution of the training set (altered codes that exceed 10 statements are discarded).

Figure 3 shows the results. Analyzing these plots reveals many observations: First, the trained model is robust to in-distribution semantics-preserving alterations; Second, if the model can successfully trace a program, it also tends to successfully trace its semantically equivalent variants produced by our five operators; Third, substantial robustness to our in-distribution semantics-preserving alterations appears in the early stages of training, while the model still has modest accuracy. Interestingly, this is true even when the performance of the base accuracy of the model is relatively low, which is especially visible in datasets 2 to 4. For example, at the first epoch in dataset 2, despite the accuracy on the base test set being at around 40%, the performance on the altered test sets reaches 90%. Furthermore, while the different experiments display an improvement in accuracy with more training, we notice considerable drops for the *neutral assignment* operator, especially in the more size-restricted training datasets (datasets 3 and 4). Comparable perturbations were noticed by Mirzadeh et al. (2024) when they applied a similar *neutral expression addition* to the GSM8K benchmark and found significant performance drops in state-of-the-art LLMs. However, no assurance could be brought concerning whether their alteration operator would strictly fall into the training distribution of the LLMs, contrary to our case, since neutral assignments appear in the training dataset.

**Robustness on the full test set (not on successfully traced snippets only)**   To test whether the robustness to our five semantics-preserving edits reported in the previous experiment reflects genuine invariance rather than a selection artifact, we repeat the previous experiment but without *success-only conditioning*. We instead apply the edits to every item in the held-out test set (one operator at a time as we did before). Figure 12 in Appendix E shows the results. In this success-agnostic setting, we observe that the accuracy under each edit closely matches the accuracy on the unaltered test,

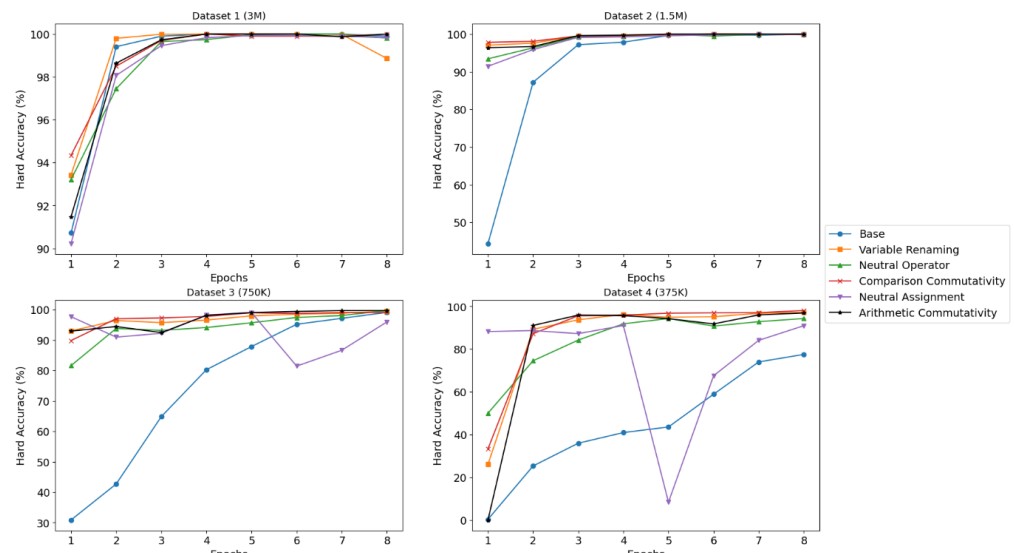

Figure 3: Robustness on successfully traced codes to in-distribution alterations

indicating that the model is robust to the edits. Because this holds without filtering to already-traced codes, the earlier robustness is not a by-product of selecting successfully traced snippets.

## 4.2 GENERALIZATION TO UNSEEN DATA SAMPLES UNDER CONTROLLED DATA COVERAGE

In this experiment, we investigate the ability of our model to handle code assignments of the form `var = var1 op var2` when a controlled subset of variable pairs is deliberately excluded from the training distribution. For example, if we exclude assignments that have $(a, b)$ on their right-hand side from the training set, can the model learn to trace code snippets that have the excluded assignments? Variable names in the right-hand side of assignments are drawn uniformly from the 26 lowercase letters, which yields a total of $26 \times 27/2 = 351$ distinct unordered pairs of variables (we consider the pair $(a, b)$ to be identical to $(b, a)$). If the pair $(a, b)$ is excluded, then assignments such as `x = a + b` or `y = b - a` will never appear in the training data. Unordered pairs are taken with replacement, which means that pairs of the form $(v, v)$ exist (e.g., $(a, a)$).

To enforce this systematically, we construct training datasets of 3 million TinyTracing snippets while forbidding a fixed percentage of the 351 possible pairs. The exclusion percentages range over $\{10, 20, 30, ..., 90, 95, 96, 97, 98, 99, 100\}$. For example, in the 10% setting, we remove 35 pairs, so the model can still observe 316 different variable pair combinations. At the extreme, in the 100% setting, all 351 pairs are excluded, meaning the training data contains no assignments of the form `var = var1 op var2`.

**Evaluation and test sets.** For evaluation, we generate two test sets of 1,024 snippets each. The first mirrors the training distribution and contains only allowed pairs; it is used to verify that the model has properly learned from its training data. The second is constructed so that every snippet includes at least one excluded pair, directly testing the model's generalization to unseen combinations. In the test set containing the excluded pairs, we ensure that the assignment containing the excluded pair is guaranteed to be executed during program flow (i.e., we make sure it will be executed if the program has if-conditionals).

**Results and analysis.** Figure 4a summarizes the model's performance across the different exclusion percentages. Performance on the Allowed Pairs Test remains consistently high across all percentages, confirming that the model has successfully learned from the training data regardless of the percentage of excluded variable pairs. This rules out underfitting as a source of error and ensures that any observed degradation is specific to the excluded combinations.

For the test set containing excluded variable pairs, the model demonstrates strong generalization even under limited data coverage. Accuracy remains above 80% for exclusion percentages up to 95%, indicating that the model can successfully handle unseen variable combinations even when it was trained on a training set that has only 5% of the possible pairs. Performance begins to decline at 98% exclusion, and when all variable pairs are excluded from training (100%), accuracy collapses. This setting corresponds to a regime in which the model has never encountered a single `var = var1 op var2` assignment during training, highlighting the fundamental limitations of extrapolating to unseen patterns.

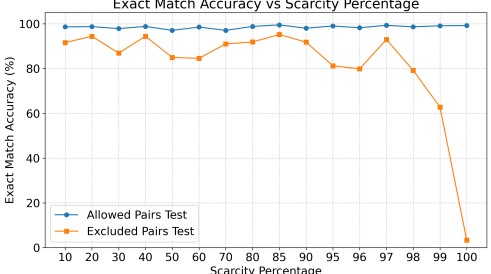 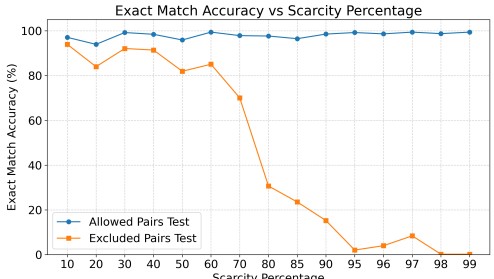

(a) Exact Match Accuracy as a function of exclusion percentage. "Allowed Pairs Test" corresponds to the test set containing only permitted variable pairs, while "Excluded Pairs Test" corresponds to the test set containing at least one assignment with an excluded variable pair.

(b) Exact Match Accuracy under global exclusion. "Allowed Pairs Test" uses only permitted pairs, while "Excluded Pairs Test" requires each snippet to contain an assignment with a forbidden pair. In contrast to Figure 4a, removing pairs globally eliminates indirect exposure and causes generalization to fail.

Figure 4: Accuracy as a function of exclusion percentage.

These results suggest that even a limited coverage of variable pairs, whether directly in assignments or indirectly in other program contexts, is sufficient for the model to generalize beyond its training distribution. However, when all occurrences of a construct are absent, performance breaks down sharply, underlining the dependence of generalization on some form of exposure.

### 4.2.1 EXCLUSION ACROSS ALL CONSTRUCTS

In the previous setup, exclusions applied only to the right-hand side of assignments. For instance, if the pair $(a, b)$ was excluded, then training snippets would never contain assignments such as case **(1)** in Figure 5, but the same

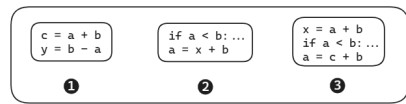

Figure 5: Exclusion regimes.

pair could still appear elsewhere in the program, for example, in conditionals or one of the variable appears on the left-hand side of assignments while the other on the right as in case **(2)**. This means that although $(a, b)$ was never observed in the restricted assignment form, the model still received indirect exposure to the pair in other contexts.

In the stricter regime introduced here, exclusions are enforced *globally*. Once a pair is excluded, it is removed from *all* syntactic constructs in the training set. Continuing the same example, if $(a, b)$ is excluded, then none of the constructs illustrated in case **(3)** of Figure 5 would appear. As a result, the model never encounters the excluded pair, removing the possibility of indirect learning.

**Results and analysis.** Figure 4b presents the exact match accuracy under global exclusion. Accuracy on the Allowed Pairs Test remains consistently high. However, performance on the Excluded Pairs Test now deteriorates earlier as exclusion percentages increase (compared to the previous experiment). Unlike the earlier setting where accuracy stayed above 80% even when 98% of pairs are excluded, here the accuracy of the model drops below 80% when 60% or more of the pairs are excluded globally. This contrast shows that the strong generalization observed previously relied on indirect exposure to variable pairs in other roles. When coverage is removed globally, the inductive bias of the model alone is insufficient, and extrapolation to unseen combinations diminishes.

### 4.3 GENERALIZATION TO OUT-OF-DISTRIBUTION SAMPLES

Our third study evaluates the model's out-of-distribution generalization (OOD) (we use the same trained model described in Sec. 3 unless mentioned otherwise). First, we describe two new test datasets used to evaluate the model's OOD generalization performance. We then present the results.

**OOD generalization tests.** We test our trained model on OOD data to measure its ability to generalize. Before describing the

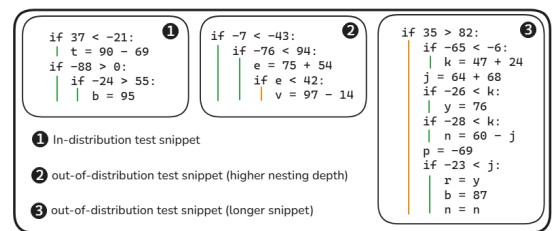

Figure 6: One ID and two OOD snippet examples

OOD perturbations, we highlight that the snippet examples of the in-distribution data are restricted to comply with two constraints: the total number of statements must not exceed 10, and the if-block nesting depth is limited to 2. Given these rules, we generate a test set that has two OOD perturbations. First, the total number of statements is between 11 and 17, and second, the nesting

depth is between 3 and 4. The goal is to test the model on snippets with lengths and depths never seen during training, justifying why such perturbations are considered OOD. All examples pass our context-window filter (pairwise steps $\leq 512$ tokens). Figure 6 shows an example, an in-distribution snippet on the left (1) and two OOD test snippets, one with a higher nesting depth and one with more statements ((2) and (3)).

**Results and analysis.** We evaluated the trained model on three test sets of 1,024 examples: the first contained in-distribution data, while the second and third included out-of-distribution examples with, respectively, deeper nesting and longer snippets (as described previously). The model achieved 100% exact match accuracy for in-distribution data, while struggling to trace any snippet with deeper nesting (0%). In this case, the model fails at reproducing the snippets correctly because it does not recognize the new nesting levels, thus clipping all lines at a depth of 3 or 4 to only 2. The model occasionally deletes if statements when their nesting depth surpasses 2. The model managed to trace correctly 43.65% of longer snippets. This difference in accuracy gives insight that OOD generalization performance does not depend only on factors related to the model but also on the nature of the distribution change. In this example, and within our evaluation settings, learning the concept of nesting depth seems harder than the concept of snippet length.

### 4.3.1 GENERALIZATION ACROSS NESTING DEPTHS.

We further investigate whether the model can generalize along the axis of *if*-statement nesting. Starting from the standard setup (maximum depth $= 2$), we expand the training distribution to include snippets with depths 0–5 and increase the maximum number of statements to 15 to allow more conditionals. We then probe generalization to deeper nesting depths by evaluating on snippets with depths 6, 7, 8, and 9.

Concretely, we generate a base training set of 3 million snippets in which $p\%$ of snippets have a depth in (6,7) and the remainder have a depth in $\{0, 1, 2, 3, 4, 5\}$. We vary $p \in \{0, 1, 5, 10, 20, 40, 60, 80, 100\}$. At test time, we evaluate on five sets of 1,024 examples: (i) a validation set sampled from snippets with maximum depth 5, and (ii) four sets containing only depths 6, 7, 8, and 9, respectively (many of these test sets are OOD).

Detailed results and plots are provided in Appendix F. The results show three consistent patterns. First, even without exposure to depths $\geq 6$, the model extrapolates to depth-6 (94.9% exact match at p=0). Here, p=0 indicates that the model is trained on snippets that have a maximum nesting depth of 5 (therefore, the model did not see depths of 6 and above). Second, introducing only a small fraction of depth-6 and 7 examples suffices to unlock generalization to both of these depths ($\geq 99\%$ accuracy once $p \geq 1$) and to unseen depth-8 snippets (up to $99.4\%$). Third, the benefit does not extend to depth-9: performance on depth-9 remains much lower, suggesting that exposure to maximum depth $X$ enables generalization to $X + 1$, but accuracy degrades beyond $X + 1$.

### 4.4 ABLATION STUDY

**Supervised Fine-tuning (SFT)** During the design of our framework, we also studied the effect of Supervised Fine-tuning (SFT) Ouyang et al. (2022) combined with Instruction Masking (Shi et al. (2024)) on the accuracy of the model in ID and OOD cases. To achieve this, we applied LoRA (Low-Rank Adaptation) Hu et al. (2021) and instruction masking on our original trained model, then evaluated the model on the same test sets used in Sec. 4.3. Results show that the model obtained shows slight but not substantial improvements in OOD generalization performance, while degrading the model's accuracy on ID data. Therefore, we decided not to use SFT and Instruction Masking in our base model. More details in Appendix (G.1).

**Relative Positional Encoding** We compare three positional encoding schemes: (i) *learned Absolute Positional Embeddings (APE) Vaswani et al. (2017)*; (ii) *Relative Positional Embeddings (RPE) Shaw et al. (2018)*; and (iii) *Relative Positional Bias (RPB) Raffel et al. (2020)*. To maintain model comparability with the APE baseline, RPE and RPB models were implemented at the same scale by adjusting only the embedding dimension while keeping layers, heads, and context window fixed. Among all of these positional encodings, APE delivered the best accuracy across all the tasks; therefore, we use it in our base model. The detailed results are presented in Appendix G.2.

## 5 RELATED WORK

**1. Robustness of language models to in-distribution perturbations.** Moradi & Samwald (2021) showed that neural models suffer performance drops under semantically neutral perturbations such

as typos or synonym substitutions. Similarly, Mirzadeh et al. (2024) tested LLMs' mathematical reasoning under modifications to numerical values or problem phrasing, revealing that accuracy degrades even when the logical structure remains. Jiang et al. (2024) highlighted that single-token changes can lead to incorrect inferences. Focusing on code, some early work explored neural models' adversarial robustness (Henkel et al., 2022; Bielik & Vechev, 2020). A recent work close to ours is that of (Orvalho & Kwiatkowska, 2025), who study semantics-preserving mutations applied to code, though using pre-trained LLMs, and find that state-of-the-art code LLMs are not robust to semantics-preserving edits. Their objective is different from ours, where we aim to control the distribution of the training data to be able to perform fine-grained experiments.

**2. Generalization of language models to out-of-distribution data.** Song et al. (2025) examined OOD generalization and its relation to composition (under synthetic settings), showing the role of induction heads in learning hidden rules. By evaluating the robustness of ChatGPT from an OOD perspective, Wang et al. (2023) showed that its performance still has limitations, suggesting that this domain is still underexplored. Yuan et al. (2023) highlights the importance of OOD benchmarks with challenging distribution shifts to accurately measure OOD performance and suggests a standardized benchmark. This aligns with our work since we can accurately control the distribution shift with synthetically generated data.

**3. Use of synthetic data and small Transformers.** Hupkes et al. (2020) generates via grammars synthetic datasets containing examples of basic string manipulation functions and studies how small Transformers handle generalization on predicting the output of these string operations when composed. (Naïr et al., 2024) explore curriculum learning by training small Transformers to predict the output of small code snippets generated using context-free grammars with increasing levels of difficulty. (Ontanon et al., 2021) leverages multiple toy tasks (duplication, cartesian product, etc.) to analyze the effect of architecture on Transformers OOD generalization. Other work focuses on testing arithmetic and symbolic tasks (McLeish et al., 2024; Qian et al., 2022; Zhang et al., 2023). Unlike classic diagnostics like SCAN and the behavioral testing framework CheckList (Lake & Baroni, 2018; Ribeiro et al., 2020), we study *code tracing* with a controllable generator to probe invariances and compositionality along symbol-coverage, length, and nesting axes. Whereas WILDS targets in-the-wild distribution shifts across real datasets (Koh et al., 2021), our synthetic setup enables precise train–test controls and minimal-exposure interventions that are hard to guarantee in natural corpora. Finally, unlike *Learning to Execute*, which trained RNNs to map programs directly to outputs (Zaremba & Sutskever, 2014), we use tiny decoder-only Transformers with *stepwise execution traces* to dissect ID robustness and OOD generalization.

Our study extends these directions by combining synthetic data generation, a code-tracing task, and tiny Transformers to systematically analyze model robustness in an accessible fashion. A more detailed discussion of related work is presented in Appendix H.

## 6 LIMITATIONS

**1) Using synthetic data.** Enables precise control of distributions but lacks real-world diversity; training only on synthetic data may miss challenging cases and limit robustness. **2) Limiting the study to tiny language models.** Focuses on small models to study semantics under constraints; findings may under-represent the capabilities of larger models. Therefore, we avoid extrapolating to large-scale models. **3) Focusing on a single architecture.** Results are specific to a decoder-only Transformer and may not extend to other architectures (e.g., encoder-decoder variants). A more detailed discussion of the limitations of this study is presented in Appendix I.

## 7 CONCLUSION

In this work, we presented a minimalist, tightly controlled framework to probe robustness in tiny decoder-only Transformers. Under this setup, we find that in-distribution brittleness to the semantics-preserving edits we study is reduced as in-distribution training data increases, and this invariance emerges early in training, even when the accuracy of the model is still modest. For compositional OOD, even limited coverage of variable-pair combinations (in expressions of the form `var = var1 op var2`) yields high accuracy on unseen pairs. OOD behavior is shift-specific: the same model partially extrapolates to longer sequences but fails on deeper nesting, and in this setup, data scaling alone was not sufficient to yield generalization to deeper nesting. While our conclusions are limited to small models and a synthetic code-tracing task, the framework illustrates, in this setting, when data scaling helps, where it falls short, and characterizes a graded dependence on coverage, motivating broader controlled studies across architectures, capacities, and OOD axes.

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

## A EXAMPLE OF FULL EXECUTION TRACE

We show an example of a full execution trace in Figure 7.

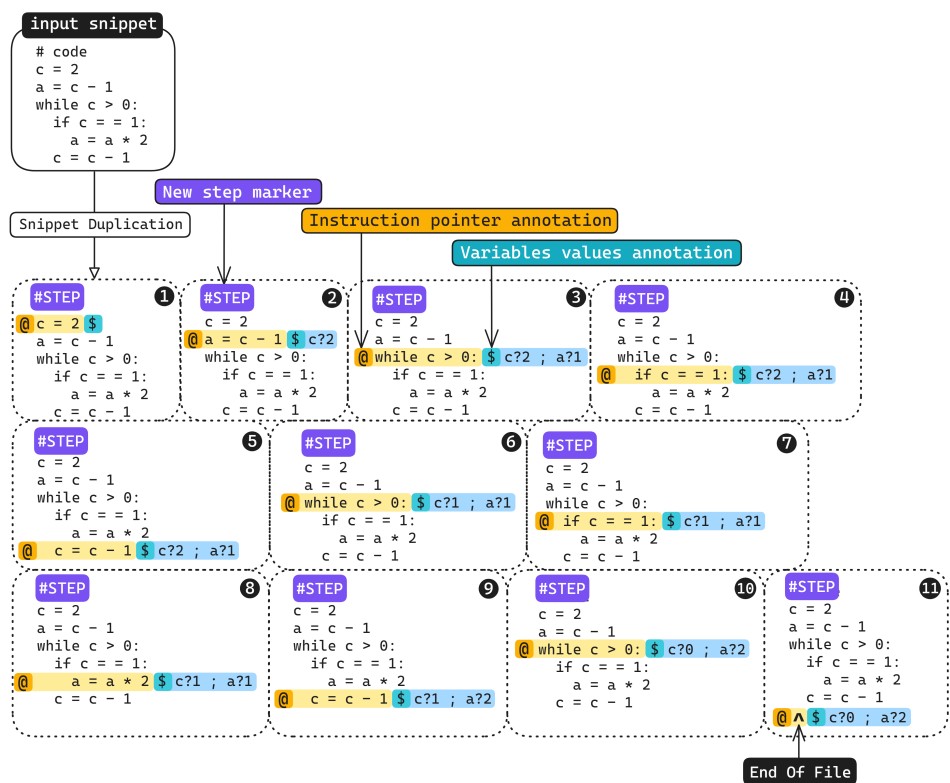

Figure 7: Illustration of the TinyTracing task. At each execution step (from 1 to 11), the entire input code snippet is duplicated and annotated with corresponding execution information: The instruction pointer value is represented by setting a special symbol (here @) to the left of the current line to be executed, and the variable states are presented as (key, value) pairs to the right of the current execution line. A special symbol (here #STEP) indicates the beginning of a new step.

## B PYTHON SNIPPETS GENERATION

This section describes the full TinyTracing code generator for completeness, including constructs such as while-loops and basic data structures. However, all experiments in the main paper instantiate a restricted configuration that emits only programs with assignments and conditionals over scalars (no loops, no recursion, and no compound data structures). Consequently, tokens and examples involving loops or data structures may appear in illustrative vocabulary tables, but they are never sampled in our training, validation, or test sets. This separation lets us present a unified generator while evaluating a simpler language necessary for our study.

## B.1 Preliminary Concepts: Skeleton Construction, Code Instantiation, and the Context Stack

As illustrated on the leftmost side of Figure 2, the snippet generator is structured around two distinct yet complementary processes: skeleton construction and code instantiation. These processes do not operate sequentially; rather, they function in tandem to produce randomly generated Python code snippets that adhere to user-specified distributional properties.

To provide context, we begin with a high-level overview of the respective roles of these two processes in snippet generation. The skeleton construction process, as the name implies, generates an abstract structure or "skeleton" of the code using a high-level intermediate language composed of specific **user-defined keywords**. In parallel, the code instantiation process consumes these keywords as they are produced, translating each into a corresponding fragment of concrete Python code. Both processes are governed by a set of **user-defined generation rules**: **high-level** generation rules control skeleton construction, while **low-level** generation rules govern code instantiation. These are illustrated in blue in the top left corner of Figure 2, representing the user input. It is precisely this parameterization via user-defined generation rules that enables the generator to produce code snippets with the desired distributional characteristics for an experiment.

To further elaborate on the internal organization of the generation system underlying the snippet generator, we now introduce a simple illustrative example. As previously noted, the set of keywords used in skeleton construction—referred to as the keyword vocabulary—is defined by the user. In the example considered here, the keyword vocabulary consists of three distinct keywords, which we describe in detail in Table 1.

As shown in Table 1, the three keywords used in our illustrative example are well defined. However, to further develop the description of the generation system of the snippet generator, we need to distinguish a special type of keywords from the others: the **context creation keywords**. These are keywords that lead to the creation of a new **execution context** in a code snippet, which is represented by **indentation** in the case of python (as an illustrative comparison, it is often the case in other languages, such as C or Java, that new execution contexts are represented with curly braces {}). Based on this definition, two such keywords exist in our example from Table 1: The **[IF_STATEMENT]** and the **[WHILE_LOOP]** (Note that while our tool allows for generating "while" loops, we do not include them in the experimental setup presented in Sec. 3).

| Skeleton Keyword | Description | Instantiation Example |
|---|---|---|
| **[ASSIGNMENT]** | Corresponds to the initialization of a random variable name with a random constant. The variable name can be one of "a", "b", or "c". The constant can be between 0 and 9. | a = 2 |
| **[IF_STATEMENT]** | Corresponds to the conditional header of an "if" block. The condition is of the form $n < m$ where $n$ and $m$ are random numbers between 0 and 9. | if 1 < 3 : |
| **[WHILE_LOOP]** | Corresponds to the looping header of a "while" block. The expression starts with "c = 0" where "c" is the control variable of the loop. Then comes the looping condition, which is of the form "c < n" where n is a random number between 0 and 9. Then directly under it and inside the loop is the update expression of the loop's control variable, which is of the form "c = c + 1". | c = 0
while c < 3 :
    c = c + 1 |

Table 1: Illustrative example of a user-defined keyword vocabulary. Each keyword is given a description along with an instantiation example.

The motivation for distinguishing context creation keywords lies in the fact that many valuable experimental studies can be conducted by manipulating distributional features specifically related to execution contexts creation. These features may include, for instance, the maximum number of

nested execution contexts, the permitted nesting combinations, or the maximum number of execution contexts allowed at a given depth. To enable controlled manipulation of such features—while maintaining the flexibility and generality of the data generator—we introduced a specialized data structure that plays a central role in the snippet generator: the **context stack**.

As its name implies, the context stack is a stack-based data structure that plays a pivotal role in the snippet synthesis process. It not only facilitates the controlled creation of execution contexts but also serves as a coordination mechanism between the skeleton construction and code instantiation processes. Broadly speaking, a new level is pushed onto the stack whenever a new execution context is introduced during generation, and correspondingly, levels are popped from the top of the stack as these contexts are closed. Each level of the context stack is itself a composite data structure, consisting of two distinct sub-components which are associated with the execution context that the level corresponds to:

- **A General Information Dictionary (GID).** This component is an extensible dictionary of user-defined **(key, value)** pairs designed to store relevant metadata about the associated execution context. Such metadata may include, for example, the type of execution context (e.g., loop, conditional), or metrics such as the number of code lines contained within the context. The GID allows for flexible customization and tracking of context-specific properties during snippet generation. These properties can then be used to further control the generation.
- **A Context Keyword Queue (CKQ).** This component is a simple queue data structure responsible for storing the sequence of keywords that will constitute the sub-skeleton of the corresponding execution context. Typically, the CKQ is populated by the skeleton construction process and later consumed by the code instantiation process.

Figure 8 illustrates the concepts outlined above on the structure and function of the context stack.

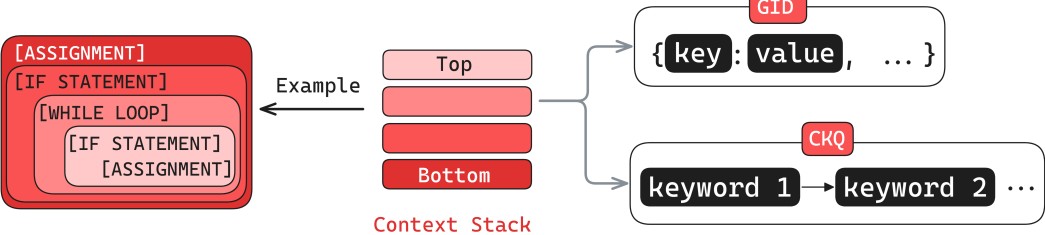

Figure 8: Diagram illustrating a context stack with four nested levels (center), including a detailed view of the substructures within a single stack level (right), and an example of a corresponding skeleton that could give rise to such a stack configuration (left).

### B.2 THE TINYTRACE-GENERATOR GENERATOR INTERFACE

Having established the foundational concepts of skeleton construction, code instantiation, and the context stack, we can now present a comprehensive description of the underlying generation system of the snippet generator.

The skeleton construction and code instantiation processes are each implemented as separate algorithms: the **Skeleton Construction Algorithm (SCA)** and the **Code Instantiation Algorithm (CIA)**, respectively. Within these algorithms, users are to define the generation rules that encode the data constraints required for their specific experiments: high-level generation rules are expressed within the skeleton construction algorithm, while low-level generation rules are defined in the code instantiation algorithm.

However, to ensure compatibility with the overall generation system, including integration with the context stack, these generation rules must conform to a predefined algorithmic template. This template, referred to as the **TinyTrace-Generator Interface (TGI)**, is illustrated in Figure 9.

As shown in Figure 9, the TinyTrace Generator Interface consists of three core algorithms that collectively define the generation system of the snippet generator. These three algorithms include:

1. The skeleton construction and code instantiation algorithms that we described previously.

2. A central **Main Algorithm (MA)** which is in charge of coordinating them.

By appropriately modifying the sections of the TGI designated for user-defined generation rules within the SCA and the CIA—highlighted with a **star (*)** symbol in Figure 9—users can implement a wide range of data distributions tailored to the requirements of their experiments. To demonstrate how this flexibility is achieved, the following paragraphs will describe the structure and functionality of each algorithm comprised in the interface, starting from the topmost component in Figure 9, which represents the main algorithm, before moving to the skeleton construction and code instantiation algorithms.

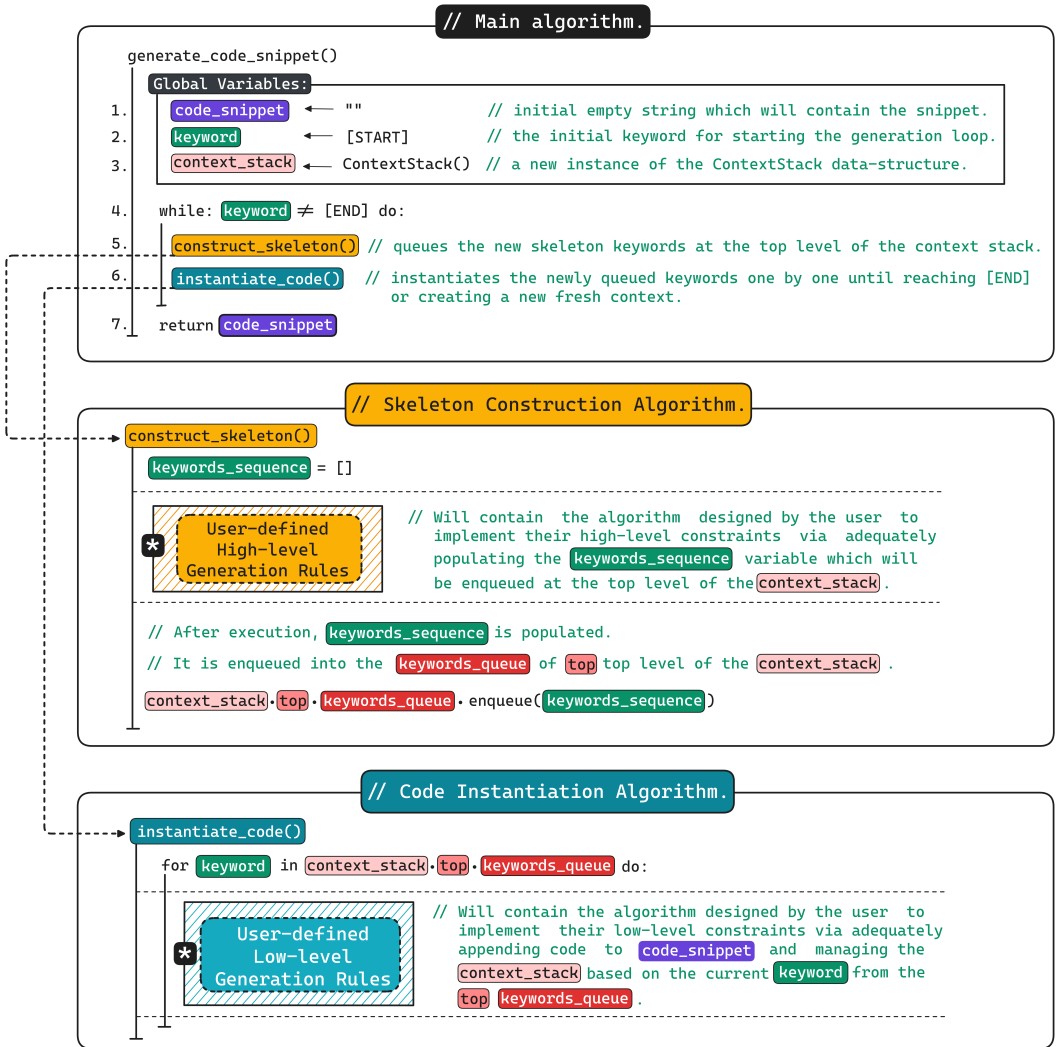

Figure 9: The TinyTrace Generator Interface (TGI) of the snippet generator.

**Algorithm 1.** **The Main Algorithm.** This algorithm is implemented by the **generate_code_snippet()** function. It begins by initializing three global variables that are central to the functioning of the generation system: **code_snippet**, **keyword**, and **context_stack**. Understanding the role of each of these variables provides insight into the overall logic of the main algorithm and offers a preliminary view of the two other auxiliary algorithms, which it coordinates:

- **code_snippet**: A string variable that accumulates the final generated code. It is initialized as an empty string, which is to be progressively extended with new code fragments as they are generated by instantiation.
- **keyword**: This variable holds the current skeleton keyword to be instantiated into a Python code fragment. It is initialized with the special keyword **[START]** to enter the main generation loop (line 4) and is to be eventually set to **[END]** to signal the termination of snippet generation.
- **context_stack**: This variable holds the context stack, which we described earlier and depicted in Figure 8. It is initialized with a single level representing the outermost execution context, which is also called indentation level 0.

Following these initializations, the main generation loop (line 4) is entered. The loop's structure is deliberately simple: at each iteration, it invokes the **construct_skeleton()** procedure (line 5), followed by the **instantiate_code()** procedure (line 6). These two procedures correspond to the **Skeleton Construction** and **Code Instantiation** algorithms, respectively, and will be described in detail in the following dedicated paragraphs.

Broadly speaking, **construct_skeleton()** is responsible for generating new skeleton keywords—according to user-defined rules—and queuing them in the topmost level of the context stack (i.e., the **current context**). These keywords are then sequentially dequeued and processed by **instantiate_code()**, which translates each into a corresponding Python code fragment. This translation is also governed by user-defined rules and may involve additional actions necessary to maintain consistency with the overall generation system. Once the generation loop is exited, the final **code_snippet** is returned (line 7).

**Algorithm 2.** **The Skeleton Construction Algorithm.** This algorithm is implemented by the **construct_skeleton()** procedure. The algorithm begins by initializing a local variable named **keywords_sequence**. This variable, which begins as an empty list, is to be filled with the sequence of new keywords that should be enqueued in the current context (i.e., the top of the stack). Right after this initialization is the user-defined area for the high-level generation rules, which is the portion of the interface that the user must edit in order to implement the desired high-level distributional constraints. Basically, in order to integrate consistently with the rest of the generation system modeled by the TGI, this user-defined region must populate the **keywords_sequence** variable with the appropriate keywords to be enqueued in the current context, at the end of the **construct_skeleton()** procedure, as shown by the interface.

**Algorithm 3.** **The Code Instantiation Algorithm.** This algorithm is implemented by the **instantiate_code()** procedure. The algorithm begins by entering a local main loop. This main loop will dequeue the keywords of the current context, one by one, into the global **keyword** variable. Inside the loop is the user-defined area for the low-level generation rules, which is the portion of the interface that the user must edit in order to implement the desired low-level distributional constraints. Basically, in order to integrate consistently with the rest of the generation system modeled by the TGI, this user-defined region must be implemented so that for each keyword, the corresponding code fragment is appended to the global **code_snippet** variable, and, **in case of a context creation keyword**, a new context must be pushed onto the context stack.

— Figure 10 provides an example of a TGI-consistent user implementation for the high-level and low-level generation rules, expressed in pseudo-code. These user-defined generation rules allow for generating Python snippets with the following distributional constraints:

1. **High-Level Constraints:**
   - The snippets are always structured as an if block followed by an initialization statement.
   - The interior of the if block can either be structured as a while block followed by an initialization, with a probability of 30%, or two consecutive initializations, with a probability of 70%.
   - The interior of the while loop is empty of any other constructs.

2. **Low-Level Constraints:**

- These are represented by the keyword translations described in Table 1.

As illustrated in the example of Figure 10, the user-defined generation rules take the form of an imperative description: that is, they are expressed through procedural algorithms which specify the desired structure of the code snippets in a manner that integrates coherently with the rest of the generation system imposed by the TinyTrace Generator Interface.

Figure 10: Example of a TGI-consistent implementation of the user-defined high-level and low-level generation rules.

## C    MODEL HYPERPARAMETERS AND TRAINING DETAILS

### C.1    MODEL TRAINING

We train from scratch with the AdamW optimizer using a two-phase schedule: Phase 1 runs for half of the epochs with a linear warmup over the first 10% of steps from $10^{-4}$ to a peak $10^{-3}$, followed by cosine decay back to the initial LR; Phase 2 continues for the rest of the epochs with all LRs reduced by 90% (initial $10^{-5}$, peak $10^{-4}$). Batches contain 512 examples and training uses data parallelism on $4\times$ NVIDIA A100 80GB GPUs. During training, mini-batches are randomly sampled at a fixed context length $T{=}512$. Decoding at evaluation is greedy; the metric is strict exact-match including all tracing markers (e.g., #STEP, @, $..., ^) across the full generated trace.

### C.2    TOKENIZATION

We use a custom tokenizer that has a fixed 77-token vocabulary tailored to the TinyTracing format, enumerating the task's lexemes (e.g., lower-case identifiers a–z; control keywords if/elif/else/while; arithmetic/comparison operators $\{+, -, *, //, \%, <, >, \leq, \geq, ==, !=\}$; assignment/punctuation; digits), together with the tracing markers (@, $, ^, #STEP and newline). Integers are tokenized at the character level: each decimal digit is a separate token, and the minus sign (for negatives) is a separate token (e.g., -37 $\rightarrow$ '-', '3', '7'). This design keeps all symbols atomic (no learned subwords).

Although the tokenizer contains a superset of operators, in this paper's datasets, we only use $\{+, -\}$ for arithmetic and $\{<, >\}$ for comparisons; the other operator tokens are not used.

## C.3   MODEL HYPERPARAMETERS

Table 2 shows the hyperparameters we used when training our model.

| Parameter | Value |
| --- | --- |
| Model Type | Decoder-only Transformer |
| Number of Layers | 12 |
| Number of Heads | 16 |
| Embedding Dimension | 368 |
| Context Window | 512 tokens |
| Total Parameters | $\sim$20 Million |
| Normalization | RMSNorm |
| FFN Activation | SiLU |
| Positional Encoding | Learned Absolute Positional Encoding |

Table 2: Key hyperparameters of the probed Transformer model.

## C.4   DATA DEDUPLICATION

To avoid duplicates, we enforce uniqueness during data generation: once a code snippet is generated, it is hashed to produce a unique ID; if that ID has already appeared, the snippet is discarded and not added to the dataset. For hashing, we use SHA-256 over the raw snippets. Deduplication is enforced on the training, evaluation, and test sets.

To prevent leakage, we enforce *global* deduplication across train/validation/test. Concretely, we maintain a single global hash set while constructing the corpus: any example whose hash is already present is discarded *before* splitting. This procedure ensures both within-split uniqueness and cross-split disjointness.

## C.5   DETERMINISM AT EVALUATION VS. WINDOWED TRAINING.

In Sec. 2.2, we require that two successive steps in the trace lie within the model's context window to preserve *deterministic* execution during *evaluation* (i.e., the next step is a function of the visible state without truncation effects). By contrast, during pre-training we use standard *windowed sampling*: random contiguous 512-token spans drawn from the corpus, which may cut across step boundaries and do not enforce the step-pair constraint. This separation ensures (i) faithful, deterministic evaluation and (ii) efficient, unbiased pre-training.

## D   EXAMPLES OF IN-DISTRIBUTION SEMANTICS-PRESERVING ALTERATIONS

Figure 11 shows examples of our in-distribution semantics-preserving alterations.

## E   ROBUSTNESS TO IN-DISTRIBUTION SEMANTICS-PRESERVING ALTERATIONS ON THE FULL TEST SET (NOT ON SUCCESSFULLY TRACED SNIPPETS ONLY)

Figure 12 shows the results of the experiment.

## F   GENERALIZATION ACROSS NESTING DEPTHS

We describe here the experimental setup and detailed results for our study on generalization across nesting depths. By "nesting depth," we refer to the maximum number of nested if-blocks or in-dentation levels in a Python snippet; for example, a snippet with a single if inside another if has a depth of 2. The initial training corpus contained 3M snippets with maximum depths ranging from 0 to 5. For each percentage $p \in 0, 1, 5, 10, 20, 40, 60, 80, 100$, we constructed a new training set by

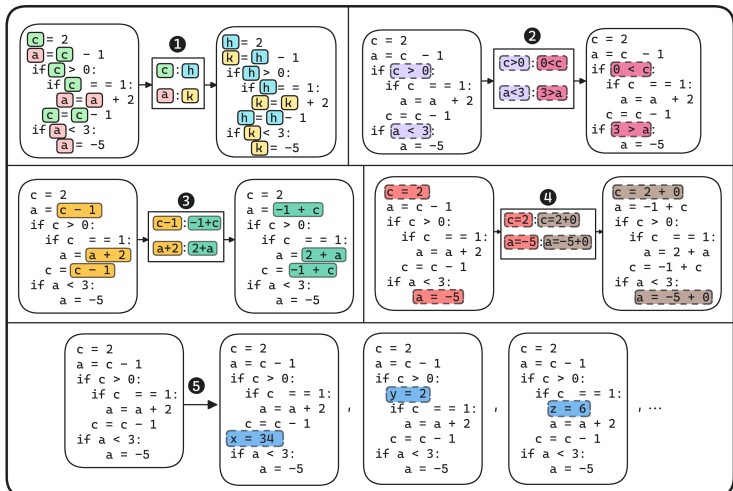

Figure 11: Our in-distribution alteration operators (1–5). **(1) Variable renaming**: variable identifiers are replaced with other valid identifiers from the same distribution, preserving semantics. **(2) Comparison symmetry**: conditional comparisons are made equivalent by simultaneously swapping operands and flipping to the symmetric comparator (e.g., $a < b \mapsto b > a$). **(3) Addition commutativity**: swap the order of operands in addition expressions (e.g., $a + b \mapsto b + a$). This transformation is applied exclusively to addition, since subtraction is not commutative. **(4) Neutral operator**: a $+0$ or $-0$ is inserted in an assignment (e.g., $x = y \mapsto x = y + 0$), which preserves semantics; note that 0 naturally occurs in the training data range $[-99, 99]$, so such edits are in-distribution. **(5) Neutral assignment**: a new assignment is inserted at a random location, defining a fresh variable name not used elsewhere, ensuring the execution trace is unaffected.

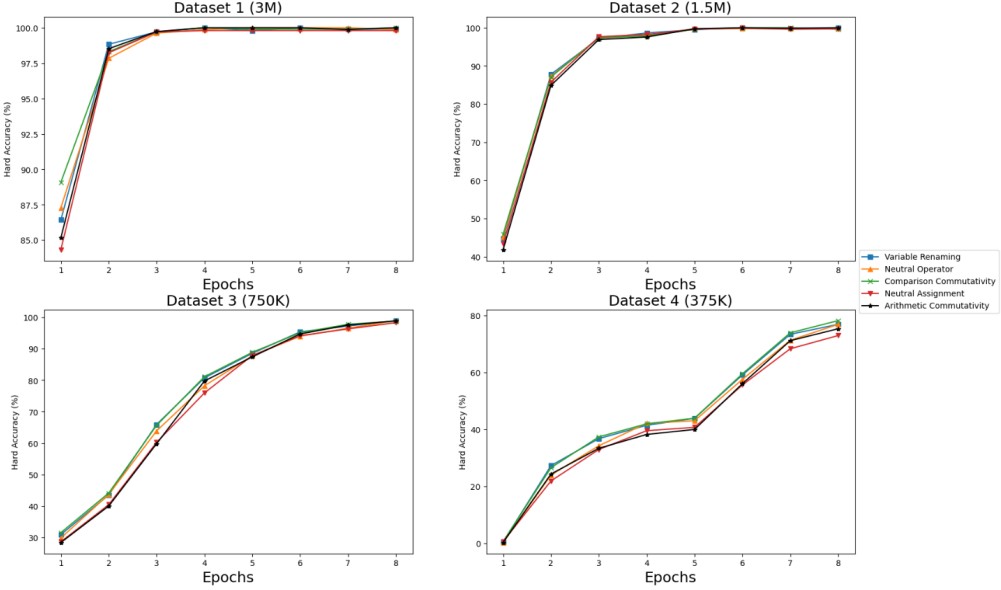

Figure 12: Robustness on the full test set to in-distribution alterations

keeping $(100 - p)\%$ of the original dataset and injecting $(p/2)\%$ of depth-6 and $(p/2)\%$ of depth-7 examples. This allows us to study the effect of gradually introducing deeper nesting patterns.

At evaluation time, we used five test sets of 1,024 examples each. The first was an in-distribution (ID) set, sampled using the same technique as the training data. The remaining four sets were maximum-depth test sets, each containing snippets in which the maximum nesting depth reaches 6,

7, 8, and 9 at least once. Nesting depths 6 and 7 were gradually introduced into the training data as described above, so these sets became partially in-distribution at higher training percentages. In contrast, nesting depths 8 and 9 were never included in training and thus remained fully out-of-distribution (OOD) throughout the experiment. We reported the exact match accuracy for all test sets.

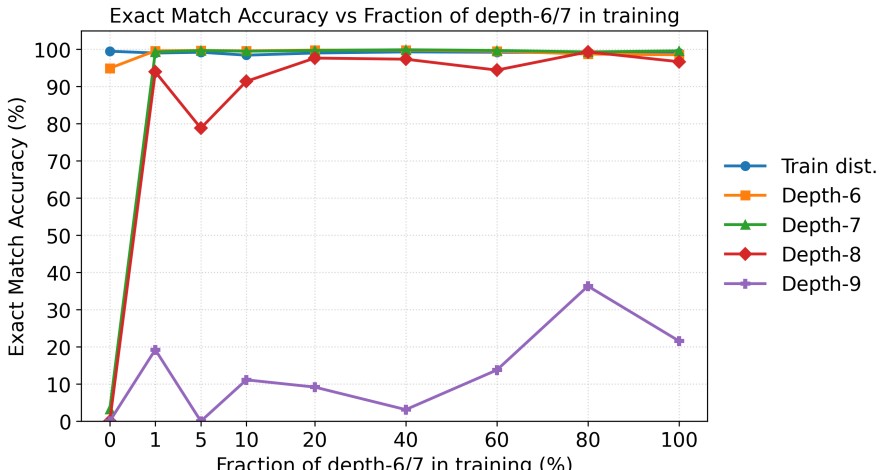

Figure 13: Exact-match accuracy (%) as a function of the fraction of nesting depths 6 and 7 snippets included in training. Curves correspond to evaluation sets with different maximum depths: the training distribution ("labeled Train dist.") and held-out deeper snippets ("Depth-6", "Depth-7", "Depth-8", "Depth-9"). The plot illustrates three key trends: (i) zero-shot generalization to the immediately deeper depth (Depth-6) even when no such examples are present in training ($p = 0\%$), (ii) rapid stabilization of accuracy for depths 6–8 with minimal exposure ($p \geq 1\%$), and (iii) persistent degradation for Depth-9 across all fractions, highlighting limits of generalization to much deeper structures.

The results show that when a model is trained on a dataset with maximum depth $X$, it generalizes reliably to $X + 1$ nesting depths without explicit exposure. For example, depth-6 accuracy is high even when the training set contains only depths up to 5, and similarly, depth-8 accuracy improves once depth-7 examples are included. Moreover, introducing even a small fraction of deeper examples (e.g., 1%) is sufficient to extend robust generalization to higher depths, as seen for depths 6 and 7. However, exact-match performance drops sharply at depth-9 and for higher depths when the maximum exposure in training is limited (e.g., $X = 5$), indicating that generalization rarely extends beyond $X + 2$.

## G EXTENDED ABLATION STUDY

### G.1 SUPERVISED FINE-TUNING (SFT) AND INSTRUCTION MASKING FOR OOD GENERALIZATION

In addition to evaluating the current model's out-of-distribution generalization capabilities, we explored whether these can be improved using SFT and Instruction Masking. In this section, we explain technical implementation details related to the pre-evaluation phase (fine-tuning the pre-trained model), presenting in the process the hyperparameters at play when running the experiments, followed by the best results we obtained.

**Supervised fine-tuning explained.** After training our model, we use LoRA (Low-Rank Adaptation) to fine-tune it Hu et al. (2021). The data used for fine-tuning belongs to the same distribution as that used to train the model. The difference lies in a technique called *instruction masking*, which is described as follows: for each example in a batch, the model sees two parts, an *input* and an *output*. Each token of the example belongs to one of these parts. The model initially attempts to

predict every token of the example based on the preceding tokens, but it only learns from tokens in the *output* region. This is achieved by masking the logits generated after predicting tokens in the *input* region.

**Hyperparameters and Results.** We conducted several fine-tuning experiments on the trained model described in Sec. 3, varying both data size (ranging from 65,536 to 2,097,152 examples) and the number of epochs (between 1 and 8) we used AdamW with $\beta_1 = 0.9$, $\beta_2 = 0.95$, and a weight decay of 0.1. The initial learning rate was set to $1 \times 10^{-3}$ with cosine decay down to $10\%$ of its initial value, and a linear warmup over the first $10\%$ of training steps. Fine-tuning was run with a global batch size of 512 sequences (context length 512 tokens). Evaluation was performed every $2.5\%$ of fine-tuning steps on a held-out validation set, with checkpoints saved for the best validation loss. We used LoRA rank 8, with trainable low-rank adapters replacing linear layers and all other parameters frozen except LoRA weights and biases (reducing the number of trainable parameters from 20 million to 2.2 million). We then evaluated each model—including the pure trained model for comparison purposes—on three test sets of 1,024 examples: the first contained in-distribution data, while the second and third included out-of-distribution examples with, respectively, deeper nesting and longer snippets. We present in Table 3 the corresponding evaluations for the pre-trained model and the best model obtained from fine-tuning (this model version was fine-tuned on one million examples, with the number of epochs set to 1).

Table 3: Evaluation results for the pre-trained and fine-tuned models (targeted distribution shifts : deeper nesting and longer sequence)

| Case | ID Accuracy | OOD deeper nesting | OOD longer sequence |
|---|---|---|---|
| Pre-trained | 100% | 0.00% | 43.65% |
| Fine-tuned | 99.22% | 0.00% | 46.68% |

We observe from Table 3 that the supervised fine-tuning technique slightly improved the out-of-distribution generalization accuracy by just under 3% only for the task of tracing longer snippets, while slightly degrading the pre-trained model's performance on in-distribution data. This trade-off suggests that supervised fine-tuning does not bring substantial improvements to the model's out-of-distribution generalization capabilities; therefore, we chose not to use this technique in our proposed model.

## G.2 ABLATION STUDY ON POSITIONAL ENCODING

| | In-Distribution | Longer Snippets | Higher Nesting Depth |
|---|---|---|---|
| Absolute Positional Encoding | 100% | 43.65% | 0.00% |
| Relative Positional Encoding | 82.62% | 12.99% | 0.00% |
| Relative Positional Bias | 100% | 24.80% | 0.00% |

Table 4: Accuracy of TinyTracing models trained with different positional encoding strategies. We compare Absolute Positional Encoding, Relative Positional Encoding, and Relative Positional Bias after 4 epochs of training. Models are evaluated on three test sets: In-Distribution, Longer Snippets, and Higher Nesting Depth.This ablation study guides the choice of the base architecture for the experiments presented in Sec. 4.

## H MORE DETAILED RELATED WORK

**Compositional generalization diagnostics (SCAN) Lake & Baroni (2018)**. A foundational line of work measures systematic generalization with controlled, synthetic tasks. The SCAN benchmark of Lake & Baroni tests whether models can recombine primitives ("jump," "twice," etc.) into held-out compositions; standard sequence models typically fail when generalization requires systematic composition rather than local interpolation (e.g., jump twice). Our TinyTracing setup plays a similar diagnostic role—but in the code domain and with tight control over structural axes (symbol coverage, length, nesting), letting us probe which axes are amenable to data scaling and which are not. Proceedings of Machine Learning Research

**Behavioral testing/invariance checks (CheckList) Ribeiro et al. (2020)**. Beyond benchmark accuracy, Ribeiro et al.'s CheckList formalizes behavioral testing for NLP through capability matrices and templated test types, surfacing brittleness to meaning-preserving edits. Our in-distribution (ID) perturbation experiments instantiate an analogous philosophy for program tracing: we build tests that target invariances (renaming, commutativity, neutral insertions) and quantify when those invariances are learned—showing that, in our setting, ID brittleness largely disappears once training data covers such variants.

**Standard OOD framing (WILDS) Koh et al. (2021)**. For in-the-wild distribution shifts across subpopulations and environments, WILDS provides a unified benchmark suite and evaluation protocol. While our study deliberately uses synthetic data to isolate causal factors, our findings mirror the WILDS perspective that OOD performance can degrade sharply—and that robustness depends on the nature of the shift. In particular, we observe length extrapolation but failure on deeper nesting and on new variable names, suggesting that some axes may require explicit exposure or inductive bias rather than mere data volume.

**Code execution with sequence models (Learning to Execute) Zaremba & Sutskever (2014)**. Our code-tracing task sits in a tradition of training sequence models to execute or reason about programs. Zaremba & Sutskever's Learning to Execute showed that LSTMs can learn to map character-level programs to outputs—provided careful curricula—highlighting both the promise and pitfalls of sequence models for program semantics. We extend this trajectory by studying step-by-step execution traces with tiny Transformers, enabling controlled ID/OOD stress-tests of invariances and compositionality.

**Additional related work.** Beyond the works already discussed, pretraining and data diversity can improve robustness but do not guarantee OOD gains: pretrained Transformers are generally more robust than older architectures yet still brittle under distribution shift (Hendrycks et al., 2020), and large-scale studies report only limited OOD improvements even as ID accuracy rises (Miller et al., 2021). At the same time, small amounts of targeted counterexamples can disproportionately help models unlearn shortcuts and generalize beyond spurious cues (Tu et al., 2020; Fang et al., 2022). On the compositional side, COGS complements SCAN/CFQ/PCFG in showing large train–test gaps under syntactic/semantic recombination (Kim & Linzen, 2020). Methodologically, simple training/architecture choices (e.g., positional schemes, stopping criteria) can materially shift systematic generalization (Csordás et al., 2021), and causal LMs may learn positional information even without explicit encodings (Haviv et al., 2022). These findings contextualize our axis-dependent OOD results (length vs. depth) and our positional-encoding ablation.

## I    DETAILED DISCUSSION OF LIMITATIONS

**1. Using synthetic, manually generated data.** While the use of synthetically generated data via manually defined algorithms allows for controlling the data distributional properties for precise model behavior probing, it inevitably lacks the variety of concepts naturally present in real-world datasets—a factor known to contribute heavily to the performance of pre-trained language models. Our models, by only being trained on a synthetic distribution, may have lacked the necessary exposure to a broader variety of adversarial samples that would contribute to their robustness.

**2. Limiting the study to tiny language models.** We intentionally restrict our experiments to models within small parameter counts to explore semantic learning in resource-constrained settings. However, this constraint may under-represent capabilities that emerge in larger models, particularly those operating at scale, where generalization and abstraction mechanisms are more solid. Our results thus reflect the behaviors and limitations of this lower capacity regime, and caution should be exercised in extrapolating them to foundation-scale models.

**3. Focusing on a single architecture.** By exclusively targeting a specific decoder-only Transformer architecture, we do not explore how alternative architectures (e.g., encoder-decoder variants) might perform on the same task. These architectures may offer different mechanisms for memory and representation that could influence performance. As such, our conclusions are limited to a specific architectural family and may not extend to other model designs.

## J  FREQUENTLY ASKED QUESTIONS (FAQ)

In this section, we address some of the frequently asked questions related to our work.

**Why focus the core task on assignments and conditionals?**  We restrict the main experiments to assignments and conditionals (no loops or compound data types) to keep the task within a ∼20M-parameter capacity regime. An earlier iteration of our setup included loops; tracing such a richer language required a 60M-parameter model trained on 15B tokens for 5 days per run, making our large number of controlled experiments computationally impractical. We therefore target a simpler subset (assignments and conditionals) that a 20M-parameter model can master, while the generator still natively supports loops and basic data structures for follow-ups. We believe that such a simpler language is a better testbed that enables cost-effective experimentation for us and for the community.

**How should the positional-encoding results be interpreted?**  The ablation equalizes parameter counts by adjusting embedding dimension; thus, the observed advantage of absolute positions here should be read as task-specific rather than a universal ranking.

**Why use exact match (EM) over full traces as the primary metric?**  Given our determinism filter, each step is a function of the previous one; partial-credit metrics can mask execution-breaking errors. EM on the full trace directly reflects end-to-end execution fidelity.

**Interpreting Early Robustness**  Robustness to semantics-preserving edits emerges early in training, even when base accuracy is modest, suggesting that invariances are learned before full task mastery in this setting. This observation motivates more studies of training dynamics under controlled distributions.

**On why we use synthetic generation**  Synthetic data gives exact control over coverage and OOD axes (length, depth, symbol pairs), letting us make causal statements that opaque pretraining corpora do not afford.

**On train–test determinism**  We decouple efficient pretraining (windowed spans) from deterministic evaluation (step-pair constraint) so that reported robustness is not confounded by context truncation.

**On comparisons to large pretrained models**  Our objective is controlled diagnosis rather than leaderboard ranking; we therefore avoid cross-model comparisons whose training distributions cannot be specified.

**On interpreting length vs. depth**  Length extrapolation partially succeeds while deeper nesting fails, indicating that longer contexts do not substitute for hierarchical state tracking; this suggests curricula or architectural biases as promising directions.

**Early emergence of invariance to the five semantics-preserving edits**  Invariance to semantics-preserving edits appears early in training and persists when edits are applied to the full test, indicating that it is not a selection artifact.

**On artifacts and reproducibility**  We will release (with the camera-ready paper) the TinyTrace-Generator configuration, the whole framework code and scripts, and all training/evaluation scripts to enable exact replication and easy insertion of new variants. We avoid releasing these with the paper during review as it is hard to fully anonymize our code.

## K  LLM ASSISTANCE DISCLOSURE

We made light use of large language models (LLMs) to (i) polish grammar and phrasing in parts of the manuscript and (ii) help identify potentially relevant related work during the literature review. All modeling ideas, methodological choices, experiments, and conclusions are our own.

## L    REPRODUCIBILITY STATEMENT

We strive to make all results fully reproducible. The problem setup and method are specified in the main text, and all implementation choices, training protocols, and evaluation procedures are cross-referenced there and detailed in the appendix (hyperparameters, optimization settings, ablations, and exact model configurations). We will release (with the camera-ready paper) the full source code, datasets, and evaluation scripts, including all parameters and hyperparameters used to produce every table and figure, as well as configuration files for each experiment, random seeds, and step-by-step run commands. The repository also includes environment specifications (e.g., requirements), preprocessing utilities for data, and scripts to reproduce metrics and plots.

