# OpenReview forum: "A Minimalist Approach for Exploring Transformer Robustness to In-Distribution and Out-Of-Distribution Samples"
_ICLR.cc/2026/Conference — ICLR 2026 Conference Withdrawn Submission_

### Official Review · Reviewer_miSd · 2025-10-21

**Soundness:** 3
**Presentation:** 3
**Contribution:** 2
**Rating:** 4
**Confidence:** 3

**Summary:**

The authors propose a toy cost-effective framework to study in-distribution and out-of-distribution generalization of transformer-based language models. By training a 20M decoder-only model on a code tracing task (which they argue is task simple enough for a tiny model to learn while being rich in terms of concepts and allowing easier controllability) they reached 4 conclusions:
- Robustness to semantics-preserving in-distribution edits can be considerably improved by augmenting the size of the training dataset with in-distribution samples.
- The robustness can already occur when the model has not yet reached its peak of performance.
- Training a model on a limited set of couples of the form `(a, b)` can be enough to generalize on unseen couples.
- Their model, trained on snippets with 5-10 statements and the if-block of depth $\leq 2$ can partially generalize to snippets with 11-17 statements but fail to produce the accurate traces for test samples with depth $\geq 3$.

**Strengths:**

- The paper proposes an interesting setup to study in-distribution and out-of-distribution with relevant alterations and good experiments.

**Weaknesses:**

- It remains unclear how these results contribute to understanding the generalization of "real-world" language models which are typically multitask, substantially larger in parameter count (even 1B models), trained on more diverse data and evaluated across varied tasks beyond simple exact match setups (for instance correctness-based evaluations in mathematics or code generation). There is little to no connections to more realistic settings.
- Figure 3: It is not a good idea to only compute the *exact match* only on the proportion of snippets where the model was right in the first place. The proportion is not the same at each stage of the training (epoch) and this makes the analysis unclear. When you computed the score on the whole set as shown on Figure 12, we see that all five alterations behave similarly during the training. It might be good idea to include the `base` on that figure and keep the same colors as in Figure 3.
- This sentence :`substantial robustness to our in-distribution semantics-preserving alterations appears in the early stages of training` is difficult to understand. The results in the paper seem to show that the model is "as robust as it is accurate". "early stages" of training is not well defined.

**Questions:**

- Could you clarify how the evaluation is conducted? You mention that the model `generates the complete execution trace given the code snippet at the prompt`, does it mean that the model generates each step sequentially and the correctness is measured as the exact match across all the steps?
- Figure 3: Why is there a performance drop on **Neutral Operator** alterations when training on dataset 4? The model basically fails correctly generate the traces of almost all the code snippets it used to handle correctly. Do you have an example of a typical failure case to illustrate this behaviour?
- Figure 4a: Isn't it surprising to have a difference of 5 to 10% in terms of accuracy between `Allowed Pairs Test` and `Excluded Pairs Test`? It would have been interesting to see what happens at 3% or 5% but these percentages should be small enough so that both accuracies are very close right? Do you think using more than 3M samples can bridge `Allowed Pairs` accuracy and `Excluded Pairs` accuracy at 10%? Moreover, the `Excluded Pairs` accuracy is basically the same from 10 to 97%, is there an overfit on the `Allowed Pairs test`?
- Figure 13 should be in the main part of the paper. It shows that 1% of code snippets of depth 6 or 7 in the training dataset is enough to perform well on test snippets of such depth, and even of depth equal to 8. However,  performance unexpectedly drops at depth=9, showing an example of such a failure would be helpful.

---

### Official Review · Reviewer_tqar · 2025-10-30

**Soundness:** 2
**Presentation:** 3
**Contribution:** 2
**Rating:** 4
**Confidence:** 3

**Summary:**

This paper aims to address the difficulties in evaluating the robustness of the generalization capabilities of large language models. To this end, the authors propose an efficient research framework that involves generating data for a specially designed and controllable synthetic code-tracing task (TinyTracing) and training a tiny, decoder-only Transformer model from scratch. Using this model, the paper conducts a series of experiments to investigate the robustness of the model's generalization capabilities. The main findings include:

1. In-distribution (ID) robustness: Increasing the amount of in-distribution training data can significantly improve the model's robustness to semantics-preserving transformations (e.g., variable renaming, addition commutativity).
2. Compositional generalization (to unseen samples): The model's ability to generalize to unseen variable-pair combinations largely depends on whether these pairs have been "indirectly exposed" in the training data.
3. Out-of-distribution (OOD) generalization: OOD generalization capability is correlated with the type of distributional shift.

**Strengths:**

1. The research motivation and methodology are reasonable: Given the current "black-box" and hard-to-reproduce nature of large models, the approach of using a highly controlled, low-cost, and reproducible environment with a tiny model and synthetic data is sound.
2. The experimental framework is highly controllable: The design of the TinyTrace-Generator tool allows for fine-grained control over the data distribution (e.g., precisely excluding certain variable pairs, controlling nesting depth), which makes causal analysis possible.
3. Strong reproducibility: The paper provides extensive implementation details in the appendix regarding the model, training, and data generation, which is beneficial for follow-up research by the community.

**Weaknesses:**

1. Insufficient necessity and complexity of the core task: One of the paper's core contributions is the design of the TinyTracing task, but its necessity is not well-justified. The task is ultimately limited to a minimal Python subset containing only assignments and conditionals (without even loops). For such a simple task, the observed model behaviors (e.g., failing to generalize to 3 levels of nesting) are likely a direct consequence of the task's oversimplification, making the study's conclusions unconvincing. It is unclear why a complex "code-tracing" framework (generating traces line-by-line) is needed instead of a simpler "code-to-result" prediction task.
2. The design of the compositional generalization experiment in Sec 4.2 is oversimplified: The experiment on generalization to unseen variable pairs in Sec 4.2 only considers combinations of 26 single-letter variables, which is essentially a very small-scale problem of token co-occurrence rather than a true problem of variable generalization. In real-world programming, the vocabulary space for variable names (e.g., user_count, total_price) is nearly infinite. The conclusions drawn from using 26 letters are hardly applicable to real-world coding scenarios, which makes the experiment's persuasive power extremely low.
3. The validity of the OOD generalization conclusion in Sec 4.3 is questionable: The "generalization to X+1" pattern proposed in Sec 4.3.2 is an interesting observation, but the paper does not investigate it in depth. The experiment only tests increasing depths to 6 and 7 based on a maximum depth of 5. This "X+1" pattern is likely a coincidence caused by the shallow depths (e.g., X=5) chosen in the experimental setup. The paper lacks a systematic validation of this pattern, making the finding appear preliminary and potentially unreliable.
4. The paper provides no evidence or strong arguments to suggest that the phenomena observed on a "20M-parameter model with loop-free synthetic code" have any guiding significance for our understanding of LLMs with "hundreds of billions of parameters trained on massive real-world data."
5. The depth of the experimental findings are insufficient: The paper's main findings, such as "increasing ID data improves ID robustness," "models struggle to generalize to OOD structures (like deeper nesting)," and "generalization depends on data coverage," are almost common knowledge in the machine learning field. The controlled experiments in this paper merely reproduce these known phenomena without offering deeper insights (e.g., why the model's generalization on nesting is so poor, and how to address it from an architectural or data perspective).

**Questions:**

My main questions for the authors stem directly from the weaknesses identified above. I would appreciate the authors' response on the following points:

1. Could the authors further justify the necessity of the complex code-tracing task for revealing the insights presented? Have you considered whether a simpler code-to-output prediction task might yield similar conclusions? If not, what specific phenomena can only be observed through the step-by-step tracing paradigm?
2. The generalization experiment in Sec 4.2 is confined to single-letter variables. How do the authors expect the conclusions regarding 'indirect exposure' to transfer to a more realistic programming context with a large vocabulary of multi-token variable names? Does the current setup primarily test token co-occurrence rather than true semantic generalization over variables?
3. Regarding the 'generalization to X+1' pattern for nesting depth, how can the authors be sure this is not an artifact of the shallow depths (X=5) tested? Would this pattern hold if the model were trained on a distribution with a much larger maximum depth, for instance, X=20?
4. Several key findings, such as the positive impact of more ID data on robustness, appear to confirm well-established principles in machine learning. Could the authors clarify what they consider to be the most novel or counter-intuitive insight that their controlled framework provides, which could not have been reasonably predicted from prior knowledge?

---

### Official Review · Reviewer_Qy8x · 2025-10-31

**Soundness:** 3
**Presentation:** 3
**Contribution:** 3
**Rating:** 6
**Confidence:** 4

**Summary:**

This paper proposes an open-source framework to study the robustness of a tiny decoder-only transformer in in-distribution and out-of-distribution settings. It starts with a simple yet concept-rich task, code tracing, and designs consistent experiments to study the effect of data scaling and data coverage on generalization. Experimental results bring several interesting findings, for example, models show distinct OOD performance in different settings.

**Strengths:**

I appreciate the authors' effort to open the black box of the large language model in terms of its inherent generalization mechanisms. The authors select an appropriate task for both models and humans, the AI community, to recognize and learn from it.

This paper studies important questions like the dynamics of models via data scaling, extrapolating to unseen patterns under controlled experiments. These findings are meaningful due to the rigorous setup of the experiments.

Also, open-sourcing their code is of great help to the community.

**Weaknesses:**

Though this paper conducts detailed and controlled experiments, I am not sure to what extent the findings in this paper could inform the existing counterfactual phenomena. For example, in the beginning, the authors say modern LLMs are brittle to small semantic-invariant perturbations on easy math problems. However, this paper finds that LLMs could well generalize to unseen patterns even if they only see a partial part of the pattern.

So this is one of my biggest concerns: will the findings in tiny LLMs with tiny tasks be robust to larger models with harder tasks? Maybe extrapolation itself evolves with data and model scaling.

**Questions:**

Can you analyze the results on line 383 and figure 13 together since these experiments target the same question?

---

### Official Review · Reviewer_6hXS · 2025-11-01

**Soundness:** 3
**Presentation:** 2
**Contribution:** 2
**Rating:** 4
**Confidence:** 3

**Summary:**

This paper proposes a new method to study ID robustness and OOD generalization of transformer models with a simple and scalable synthetic task. Specifically, the paper proposes a synthetic task where the model is asked to predict stepwise execution traces given a program, and uses controlled data distribution to study the ID robustness and OOD generalization of transformers. Evaluation shows that ID robustness can be established early in training, and OOD generalization is also achievable with limited distribution of training data.

**Strengths:**

- This paper explores an important and interesting research direction.
- The evaluation has involved a large number of comprehensive experiments to study the robustness of the transformer models to out-of-distribution data.

**Weaknesses:**

- Only using a single Exact Match metric might not be optimal for this study. While accurate, exact match metric cannot distinguish cases where models generate correct reasoning while failing to deprecate the program with 100% accuracy (e.g., miss some empty lines). Using a fuzzy metric here can provide an opportunity to conduct more fine-grained analysis on the performance of transformer models.
- While the paper has included a comprehensive evaluation with multiple experiments with different settings, there’s a lack of in-depth analysis on the reason behind the generalizability behavior of transformers. For example, comparing the activation difference inside transformer models when facing in-distribution and out-of-distribution data might be able to provide a more in-depth understanding of the results.

**Questions:**

- What is the superiority of using stepwise execution traces to dissect ID robustness and OOD generalization over directly mapping programs to outputs that has been explored in the literature?

---

### Author Response · Authors · 2025-12-01
**Withdraw**

We want to thank the reviewers for their valuable comments. These comments will greatly help us in improving our paper. Given that 3/4 of the comments were not positive, we prefer to withdraw the paper, address the comments and submit again to the next conference. Your comments will really help us in improving the quality of the paper and we really want to thank you for all of the detailed comments that you have provided for the paper.

---

### Note · Authors · 2025-12-01

I have read and agree with the venue's withdrawal policy on behalf of myself and my co-authors.